# Cardiomyopathy Associated with Anti-SARS-CoV-2 Vaccination: What Do We Know?

**DOI:** 10.3390/v13122493

**Published:** 2021-12-13

**Authors:** Alfredo Parra-Lucares, Luis Toro, Sebastián Weitz-Muñoz, Cristóbal Ramos

**Affiliations:** 1Division of Critical Care Medicine, Department of Medicine, Hospital Clínico Universidad de Chile, Santiago 8380420, Chile; 2Division of Nephrology, Department of Medicine, Hospital Clínico Universidad de Chile, Santiago 8380420, Chile; 3Critical Care Unit, Clinica Las Condes, Santiago 7591046, Chile; 4Centro de Investigación Clínica Avanzada, Hospital Clínico Universidad de Chile, Santiago 8380420, Chile; 5Division of Internal Medicine, Department of Medicine, Hospital Clínico Universidad de Chile, Santiago 8380420, Chile; sebastian.weitz@ug.uchile.cl; 6Department of Radiology, Hospital Clínico Universidad de Chile, Santiago 8380420, Chile; cramos@uchile.cl

**Keywords:** myocarditis, pericarditis, vaccines, COVID-19, SARS-CoV-2, epidemiology, prognosis

## Abstract

The SARS-CoV-2 pandemic has mobilized many efforts worldwide to curb its impact on morbidity and mortality. Vaccination of the general population has resulted in the administration of more than 6,700,000,000 doses by the end of October 2021, which is the most effective method to prevent hospitalization and death. Among the adverse effects described, myocarditis and pericarditis are low-frequency events (less than 10 per 100,000 people), mainly observed with messenger RNA vaccines. The mechanisms responsible for these effects have not been specified, considering an exacerbated and uncontrolled immune response and an autoimmune response against specific cardiomyocyte proteins. This greater immunogenicity and reactogenicity is clinically manifested in a differential manner in pediatric patients, adults, and the elderly, determining specific characteristics of its presentation for each age group. It generally develops as a condition of mild to moderate severity, whose symptoms and imaging findings are self-limited, resolving favorably in days to weeks and, exceptionally, reporting deaths associated with this complication. The short- and medium-term prognosis is favorable, highlighting the lack of data on long-term evolution, which should be determined in longer follow-ups.

## 1. Introduction

The severe acute respiratory syndrome coronavirus 2 (SARS-CoV-2) pandemic has had a significant global impact, with more than 240,000,000 cases and more than 4,900,000 deaths by the end of October 2021 [1]. This disease has motivated the development of new vaccines at the international level. Currently, more than 20 vaccines are approved for clinical use in humans globally, including messenger RNA (mRNA), viral vectors, and inactivated live virus vaccines [2,3,4,5]. Many of these have been shown to be effective in controlled clinical studies to prevent infection, but especially severe coronavirus disease 2019 (COVID-19) manifestations, such as hospitalizations, admissions to critical units, and deaths [6,7,8,9].

Globally, more than 6,700,000,000 vaccines had been administered by the end of October 2021 [1], and there has been a decrease in active cases of COVID-19 in recent months, mainly in countries with the highest vaccination rate per inhabitants [1,10]. Epidemiological studies in large populations have shown that vaccines effectively reduce the rate of infection and complications associated with COVID-19 [11,12,13].

Given the high number of vaccinations in different populations and the use of different types of vaccines (both attenuated virus and mRNA), multiple adverse effects have been described in recent months, such as venous thrombosis and other viruses’ reactivation (e.g., varicella-zoster), with the vast majority being mild to moderate [14,15,16,17,18,19]. A rarely described manifestation is the cardiac compromise caused by the anti-SARS-CoV-2 vaccine, with mainly published clinical reports and some reviews that include these.

This review aims to evaluate the currently available literature on cardiac involvement associated with vaccination, its incidence, pathophysiological mechanisms, and prognosis.

## 2. Case Report

A 16-year-old male patient had no significant past medical history. Ten days after receiving the second dose of the BNT162b2 COVID-19 vaccine (Pfizer-BioNTech, New York, NY, USA; EE.UU. & Mainz, Alemania), he began to have symptoms characterized by odynophagia, dry cough, and myalgia. He consulted an outpatient physician, and symptomatic treatment associated with prednisone 20 mg daily for five days was indicated, with a partial response to therapy. Two weeks later, he had oppressive chest pain to an intensity of 10/10, associated with nausea, vomiting, and palpitations, for which he consulted the emergency department. Upon admission, he was subfebrile (37.6 °C), tachycardic (112 beats per minute), and had high blood pressure (146/99), without other significant findings on physical examination. The study was complemented with an electrocardiogram without ischemic or rhythm alterations, ultrasensitive troponin I of 65 ng/L with a rise at two hours to 147 ng/L (upper normal limit less than 11 ng/L), and negative test for respiratory virus panel (including SARS-CoV-2). The patient was hospitalized with suspected myocarditis in this context, so he was admitted to the coronary unit for monitoring and further tests. Treatment was started with intravenous ketoprofen (50 mg every 8 h) and oral colchicine (0.5 mg every 12 h), with which there was progressive improvement in chest pain. Ultrasensitive troponin I continued to rise to a maximum of 6614 ng/L on the second day after admission, with a progressive decrease to 471 ng/L on the fourth day (Figure 1 shows the progression of ultrasensitive troponin I levels).

The patient did not present electrocardiographic disorders during hospitalization. A transthoracic echocardiogram showed right and left ventricular function without segmental wall motion abnormalities, left ventricular ejection fraction of 60% (preserved ejection fraction), and valve disorders of the pericardium without effusion. Subsequently, cardiac magnetic resonance imaging was performed (Figure 2a,b), which showed a slight increase in T1 times of the inferolateral and lower walls of the basal third of the left ventricle (on average 1150 ms), without alterations in myocardial perfusion in the sequence first step. In inversion recovery sequences after the administration of intravenous gadolinium, delayed subepicardial enhancement was observed in the inferolateral and lower walls of the basal third of the left ventricle (segments 4 and 5), in addition to a mild pericardial effusion. The findings described were compatible with myocarditis. 

The diagnosis of myocarditis secondary to the administration of the anti-SARS-CoV-2 vaccine was made by adding elements of the medical condition and the findings in the laboratory tests. He was maintained with the initial symptomatic treatment and was discharged on the fifth day after admission, with ketoprofen 50 mg every 12 h orally and colchicine 0.5 mg every 12 h orally. At the outpatient check-up one month after discharge, he was in good condition with complete resolution of symptoms.

## 3. Myocarditis and Pericarditis

Myocarditis and pericarditis have been classically described as inflammatory diseases triggered by viral infections [20,21]. The usual clinical presentation develops after an episode characterized by respiratory or gastrointestinal symptoms, but other infectious agents, such as bacteria, fungi, and protozoa, may also participate [22]. The most common infectious agents are cardiotropic viruses (with higher selectivity for heart tissue), such as adenoviruses and enteroviruses. In addition, other families can cause myocardial inflammation less frequently, including parvoviruses or some of the Herpesviridae family [23]. Other viruses that cause cardiac involvement have an indirect action mediated by abnormal immune system activation, such as the human immunodeficiency virus (HIV) and influenza A and B viruses [24,25].

## 4. Mechanisms of SARS-CoV-2 Mediated Myocarditis

Observational studies have shown that SARS-CoV-2 infection can cause myocarditis in infected patients [24,26]. In epidemics of other viruses of the Coronaviridae family, such as the severe acute respiratory syndrome coronavirus 1 (SARS-CoV-1) or the Middle East respiratory syndrome-related coronavirus (MERS-CoV), no virus-mediated cardiac involvement has been reported [27]. The latter could be caused by the lower number of infected individuals in these epidemics or the lower cardiac involvement of these viruses.

The information obtained from clinical and experimental reports suggests that SARS-CoV-2 generates myocardial involvement through various mechanisms. The first is mediated by the direct infection of cardiomyocytes by cardiac tropism [28]. The virus exhibits tropism for angiotensin-converting enzyme type 2 (ACE2) [29,30], which could explain the direct cytopathic effect [31]. This tropism could explain the alterations observed in patients with COVID-19, including elevation of cardiac biomarkers such as troponins [32,33] or alterations in cardiac images compatible with myocarditis [34].

A second mechanism would be an indirect action through the activation of the cytokine cascade in the context of an exacerbated and uncontrolled systemic inflammatory response [35], with a massive response of type 1 (Th1) and type 2 (Th2) helper T cells. This mechanism is known for other viruses, such as influenza A and B [25]. In addition, infection-induced hypoxemia and respiratory compromise impact the heart, contributing to infection-induced damage. Along with this, there would be an activation of an autoimmune response against autoantigens of myocardial cells [36,37], which has not yet been identified with precision.

## 5. Epidemiology of Myocarditis Associated with Anti-SARS-CoV-2 Vaccination

Vaccine-associated cardiac involvement is rare. Reports of myocarditis and pericarditis associated with vaccines have been described, mainly for vaccination against smallpox, which was carried out with a live attenuated virus [38,39]. A study that included cohorts of soldiers vaccinated between 2001 and 2003 (*n* = 1,737,868) showed an incidence of cardiac involvement between 2.16 and 7.46 cases per 100,000 persons [39]. There were no deaths or long-term cardiac involvement in patients who presented with this adverse effect. It has also been described for influenza vaccines [38,39,40] and hepatitis B [41]. The available registries show that this complication develops more frequently in the male population, especially in young adults [42,43]. Prior to the COVID-19 pandemic, isolated cases related to vaccines designed based on mRNA have been reported [44,45].

Concerning anti-SARS-CoV-2 vaccines, myocardial involvement is an infrequent event (less than 10 per 100,000 people) [46,47]. More cases have been reported with vaccines designed based on mRNA technology compared to other types of vaccines [44,45], mainly for the most used [46,48] such as BNT162b2 (Pfizer/BioNTech) or mRNA-1273 (Moderna, Cambridge, MA, USA). 

Regarding age, there is disparate behavior between the pediatric, adult, and older adult populations. A more exacerbated immune response has been observed in children and adolescents, indicating greater immunogenicity and reactogenicity with inoculation. A higher percentage of the population vaccinated with Pfizer is described as having higher antibody titers between the ages of 12–15 years versus young people aged 16–25 years [49,50]. Likewise, these subjects showed mild to moderate symptoms more often than older people. The occurrence of myocarditis or pericarditis is rare in pediatric patients. In adults, myocarditis is more frequent, especially after the second dose, predominantly in the male population, and with controversial data regarding the onset period, given the brief follow-up of patients in the studies that reported this type of complication [51]. The preceding section is endorsed in the United States Centers for Disease Control and Prevention (CDC) report [52]. In turn, an observational cohort study in the United States with 2,392,924 individuals with at least one dose of these vaccines as of August 2021 described an incidence of 5.8 cases per million after administering the second dose, similar to results found in another observational study [46,47].

The opposite situation occurs in the older adult population, where pericarditis predominates and can be triggered after the first or second dose of vaccine is administered [53]. All of these cases were of mild to moderate severity and without a definitive diagnosis. It is important to note that this vaccination complication had probably not been recognized until it became massive since phase III studies reported fewer vaccinated subjects than was necessary to detect the first case of myocarditis [47].

In summary, the risk of developing myocarditis would be low in the vaccinated population but higher than the incidence in unvaccinated people or previous reports in the literature for other mRNA vaccines.

## 6. Pathophysiological Mechanisms of Vaccine-Induced Myocarditis

To date, the pathophysiological mechanisms responsible for cardiac involvement associated with the anti-SARS-CoV-2 vaccine have not been clarified, and there is little experimental evidence. Several direct and indirect mechanisms have also been proposed.

### 6.1. Non-Causal Association between Vaccination and Myocardial Involvement

The determination of causality between anti-SARS-CoV-2 vaccination and the development of myocarditis is complex, fundamentally based on the temporal relationship between inoculation and the development of cardiac symptoms. Given this, it is possible that after vaccination, a patient may develop myocarditis or pericarditis due to classic causes, as detailed above. Among these causes, the undiagnosed SARS-CoV-2 infection at the time of vaccination would stand out, with an average incubation time of the virus estimated at seven days [54]. This is especially relevant in areas with a high incidence of active disease.

### 6.2. Activation of Innate Immune Response Mediated by mRNA

Many SARS-CoV-2 vaccines approved for use in humans are RNA vaccines. These vaccines contain modified nucleosides encoding the SARS-CoV-2 viral capsule spike glycoprotein, which are encapsulated in lipid nanoparticles that act as delivery vehicles to transport mRNA to cells [55,56,57,58]. Once inside the cell, the mRNA of the vaccine causes the cells to synthesize the protein they encode and stimulate cellular and humoral immune responses capable of identifying and destroying the virus that expresses this viral capsule protein.

This exogenous nucleotide material can be immunogenic and stimulate an innate immune response in organisms, generating an abnormal response with the potential to affect tissues other than the target cells of the therapy. To prevent this, nucleoside modifications are made to the mRNA used to decrease this unwanted immune response [55,59]. However, in patients with a genetic predisposition, it may not be sufficient to prevent it. The activation of cells that express the Toll-like receptor (TLR) and dendritic cells exposed to mRNA can activate pro-inflammatory cascades [59,60,61], which may have effects at the myocardial level.

In some case reports of patients with clinical and laboratory findings of myocarditis associated with anti-SARS-CoV-2 vaccination, where a cardiac biopsy was performed, no histopathological evidence of myocarditis was demonstrated [62,63]. It has been proposed that this is due to sampling bias, that the lesions are heterogeneous (not continuously expressed in cardiac tissue), or that the histological manifestations are not those previously described for other causes of myocarditis. Furthermore, it is interesting that in most cases, significant alterations in autoimmune parameters observed in other pathologies were not detected, including rheumatoid factor (RF), antinuclear antibodies (ANA), or elevation of inflammatory parameters (C-reactive protein or erythrocyte sedimentation rate) 

In a clinical case reported by Muthukumar et al., a patient diagnosed with myocarditis, developed after the second administration of the anti-SARS-CoV-2 [64] vaccine, was evaluated. In this case, an exhaustive study of immunological mediators was conducted. Elevated plasma levels of interleukin-1 receptor (IL-1R) antagonist, interleukin 5 (IL-5), and interleukin 16 (IL-16) were observed, with no changes in interleukin 6 (IL-6), tumor necrosis factor (TNF), interleukin 1 beta (IL-1β), interleukin 2 (IL-2), or interferon gamma (IFNµ). This patient also had increased plasma levels of natural killer (NK) cells, which destroy infected cells and participate in the innate immune response [65,66,67]. These preliminary data suggest a role for the abnormal activation of innate immunity in the development of vaccine-associated myocardial compromise.

### 6.3. Generation of Anti-Autoantigens by Mimicry between Spike Protein and Self-Antigens

Another potentially relevant role is the molecular mimicry between the spike protein produced by the vaccine and the patient’s autoantigens, which would lead to the production of anti-autoantigens. In in vitro studies [68], anti-SARS-CoV-2 antibodies have been shown to crosstalk with human proteins, such as alpha-myosin, a structural protein of cardiomyocytes involved in myocardial muscle contraction. However, to date, it has not been shown that these antibodies can generate an autoimmune response in tissues that express these proteins, both in animal models and in patients, as we can see in Figure 3.

The presence of antibodies against self-antigens was evaluated in the clinical case described above [64]. Interestingly, autoantibodies such as anti-aquaporin 4, anti-endothelial antigen, or anti-proteolipid protein 1 were detected. These autoantibodies have been previously reported in patients with myocarditis [69] and first-degree relatives of patients with myocarditis, which supports the existence of a myocarditis mechanism mediated by autoantibody formation. However, it has not been demonstrated that these autoantibodies can cause an autoimmune response in organisms, both in the heart and other tissues, so it could only be a non-causal correlation. 

### 6.4. Effects of Sex Hormones

Because the incidence of myocarditis and pericarditis is higher in male patients, especially young adults, the role of sex hormones in the pathophysiological mechanism has been considered [70]. Regarding the association with anti-SARS-CoV-2 vaccination, few published clinical studies have shown a higher number of cases in men than in women, especially in young adults [41,46,71]. Testosterone has been observed to exhibit inhibitory effects on anti-inflammatory cells, increased activity of pro-inflammatory M1 macrophages, and increased CD4+ type 1 (Th1) T lymphocyte response [70]. In turn, estrogens have an inhibitory effect on pro-inflammatory T lymphocytes, causing a decrease in the cellular immune response. This fact explains the observation that the highest incidence of myocarditis or pericarditis in women occurs in those of postmenopausal age [72]. However, given the characteristics of the published reports (several of these coming from studies carried out in soldiers, for example) [39,73], there is a significant selection bias, so it is not yet possible to confirm whether this complication is more frequent in the male population.

## 7. Clinical Presentation and Diagnosis

Clinically, myocarditis or pericarditis manifests itself with symptoms such as chest pain or discomfort, palpitations, syncope, shortness of breath, or pain with respiratory movements, especially if they appear a few days or weeks after receiving the immunization for SARS-CoV-2. Clinical guidelines emphasize those people with a past medical history of heart/inflammatory diseases such as pericarditis, myocarditis, endocarditis, rheumatic disease, or decompensated chronic heart failure in the last six months. Although the available studies have not shown an increase in the rate of myocarditis or pericarditis in this subgroup of patients, they are more susceptible to severe complications if this adverse effect appears [74,75,76].

The certainty diagnosis was made by endomyocardial biopsy. However, this test is only performed exceptionally because of the invasiveness of the technique, with the potential to generate serious complications [77,78]. Therefore, the clinical diagnosis of myocarditis and pericarditis is based on four pillars: clinical evaluation, electrocardiogram alterations, cardiac biomarker alterations, and images. The clinical evaluation of a patient with myocarditis, chest pain, or dyspnea without pericardial rubs stands out for myocarditis, a central element in pericarditis on cardiac auscultation. Electrocardiographic alterations include delayed atrioventricular conduction (prolongation of the PR interval) [79], defects in intraventricular conduction associated with nonspecific changes in the ST segment or T wave, and the presence in some cases of paroxysmal atrial or ventricular arrhythmias. In contrast, pericarditis abnormalities include diffuse ST-segment elevation with PR segment depression. 

Cardiac biomarkers such as troponin I, troponin T, and creatine kinase membrane (CK-MB) can be elevated in myocarditis, even within the normal limit in pericarditis. Finally, in the transthoracic echocardiogram of myocarditis, a decreased left ventricular ejection fraction was observed, whereas in pericarditis, the ejection fraction was normal but pericardial effusion was observed. Cardiac magnetic resonance imaging describes late gadolinium enhancement and T2 window edema for myocarditis, pericardial inflammation, or pericardial effusion for pericarditis. 

## 8. Treatment and Prognosis of Myocarditis/Pericarditis Associated with Anti-SARS-CoV-2 Vaccination

### 8.1. Overview of Pharmacological Management

The consequences of myocarditis are transitory, and generally do not require specific management. Published data show that patients frequently receive management similar to other causes of myocarditis. This therapy is based on symptomatic management, including non-steroidal anti-inflammatory drugs (NSAIDs) [80] and colchicine [81]. In patients with a poor response, it is escalated in immunomodulatory therapies, such as glucocorticoids [82]. Finally, intravenous human immunoglobulins have been reported to have an immune-modulating effect, reducing the immune response triggered by the inoculated antigens.

### 8.2. Prognosis

To date, there is insufficient literature regarding the prognosis of acute myocarditis or pericarditis associated with anti-SARS-CoV-2 vaccination worldwide, due to the low number of reports of this condition and the short follow-up time of patients due to the recent start of the massive vaccination campaign in early 2021 [83].

The published data of patients who presented with vaccine-associated myocarditis confirmed with endomyocardial biopsies, and followed up with non-invasive study methods such as echocardiography, revealed a deterioration in global systolic function, with a transient decrease in the left ventricular ejection fraction and subsequent recovery in weeks or months. In the same way, we do not know if it is necessary to maintain pharmacological treatment, especially in patients with reduced ejection fraction, even when restoring previous cardiac function.

Regarding the clinical prognosis, myocarditis or pericarditis has low mortality in the general population (including pediatric and adult patients). However, isolated cases of death due to myocarditis have been reported after vaccination against COVID-19 in patients with a severe decrease in ventricular function and cardiogenic shock, demonstrating the absence of coronary disease and histological study with inflammatory infiltrate compatible with myocardial inflammation [84].

No deaths due to vaccination have been reported in the pediatric population. It is relevant to consider that the information available is mainly obtained by vaccination with the BNT162b2 vaccine in the population aged 12–17 years. In most cases, mild to moderate symptoms occur with spontaneous resolution of the same without affecting cardiovascular function, as described [85]. We summarize the principal findings, treatment, and prognosis in Figure 4.

## 9. Conclusions

Myocarditis/pericarditis is an infrequent complication reported with mass vaccination against SARS-CoV-2, which has been observed mainly in real-world studies, given its low incidence. These vaccines, mainly those based on the mRNA platform, have presented this adverse effect more frequently than any previously used vaccine, probably because of their greater immunogenicity and reactogenicity. The underlying mechanisms have not been clarified, probably related to the activation of an innate immune response mediated by exposure to the nucleotide sequence and the generation of autoantibodies due to similarity between the encoded spike protein sequence and autoantigens.

Clinically, it triggers a mild to moderate disease, usually days after the inoculation of a short duration of the second dose. In severe cases, it can temporarily compromise ventricular function. There would be a differential presentation between pediatric, adult, and older adult populations: in the first two myocarditis predominating, and pericarditis being more frequently observed in the latter. The aim of treatment in most cases is alleviating symptoms given their favorable evolution in the short and medium term, and without having information on the long-term effects given the short time of use of these vaccines.

It is necessary to generate new studies that go along the lines of revealing the pathophysiological aspects of this adverse effect of vaccination to establish strategies that reduce the probability of developing this type of complication. In addition, long-term studies are required to determine the prognosis, both in terms of function and survival, since mRNA technology will be increasingly recurrent in the design of new vaccines to manage various pathologies of importance in public health, both for SARS-CoV-2 and other infectious agents.

## Figures and Tables

**Figure 1 viruses-13-02493-f001:**
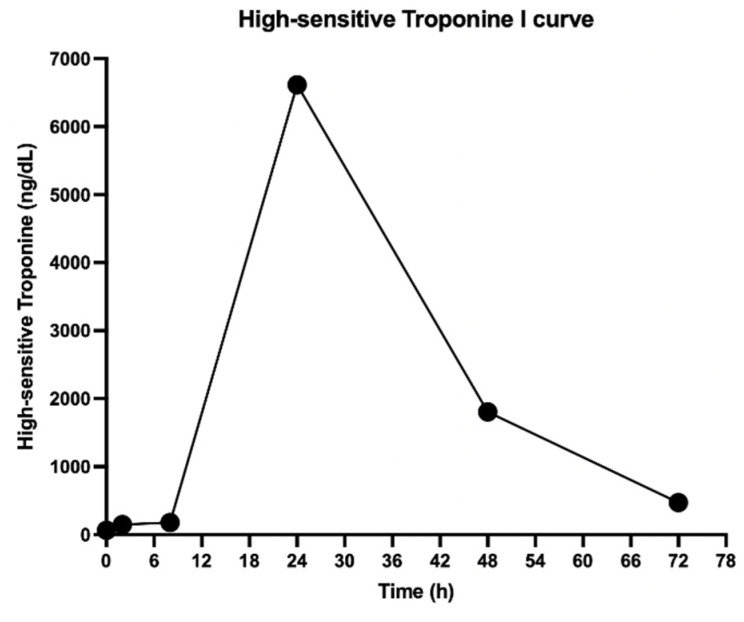
High sensitivity Troponin I curve of the patient.

**Figure 2 viruses-13-02493-f002:**
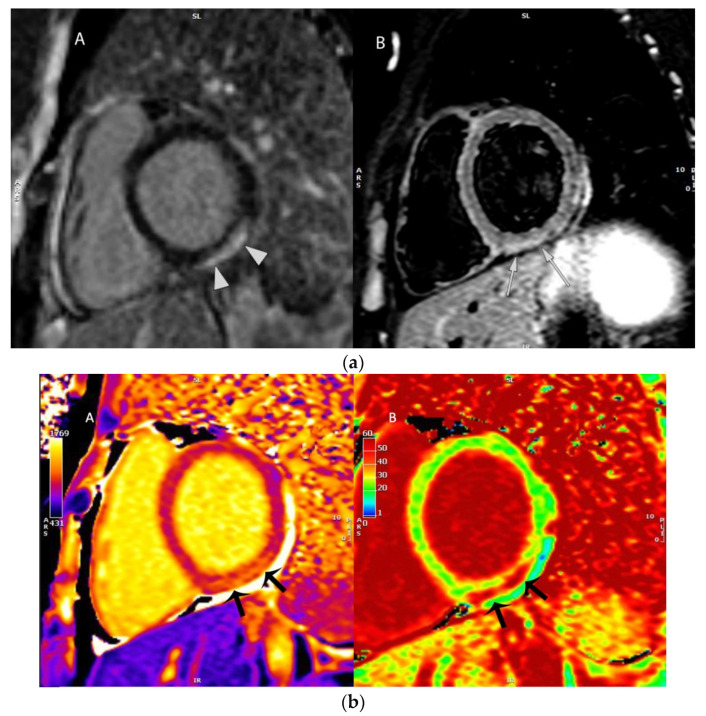
(**a**). Images of cardiac magnetic resonance (CMR). (**aA**): Phase-sensitive inversion recovery (PSIR) sequence 10 min post gadolinium shows subepicardial late enhancement of left ventricular (LV) inferior wall (arrows) and linear hyperenhancement of pericardium (arrowheads), indicating inflammatory changes related to myopericarditis. (**aB**): Short tau inversion recovery (STIR) sequence shows hyperintensity in LV inferior wall (arrows) indicating myocardial edema. Both images are orientated in the short axis plane at the LV basal level. (**b**). (**bA**): T1 map shows focal elevation of T1 times in the left ventricle (LV) inferior wall, to 1177 milliseconds, with normal values of 980 milliseconds in remote myocardium. (**bB**): Extracellular volume (ECV) map shows increment of the ECV in the LV inferior wall to 47%, with normal values of 25% in remote myocardium. Both images are orientated in the short axis plane at the LV basal level.

**Figure 3 viruses-13-02493-f003:**
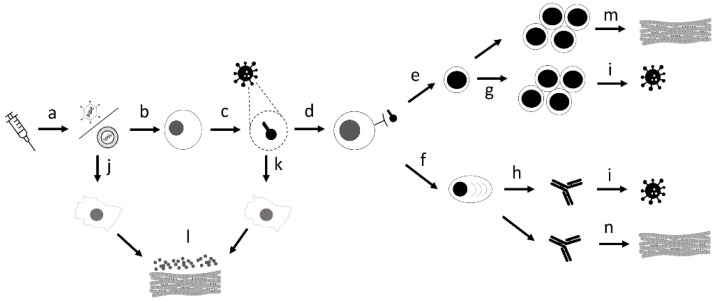
A visual summary of the proposed mechanisms related to myocarditis associated with anti-SARS-CoV-2 vaccination. (a) The host is inoculated with a vaccine containing an mRNA sequence encased in a lipid nanoparticle coat (mRNA vaccine, e.g., BNT162b2 Pfizer-BioNTech, mRNA-1273 Moderna) or a DNA sequence encased in a virus vector capsid (replication-defective viral vector vaccine, e.g., AZD1222–Oxford/AstraZeneca {Cambridge, Inglaterra}, Ad5-nCoV CanSino Biological {Tianjin, China}). This sequence codifies a specific coronavirus spike protein. (b) The nucleic acid enters the host cell and (c) translates the sequence to a coronavirus spike protein. (d) The spike peptides are presented by antigen-presenting cells (APCs), which activate adaptative immunity, including (e) cellular and (f) humoral response. (g) The cellular response includes activation of virus-specific CD4+ and CD8+ lymphocyte T cells, and (h) the humoral response includes activation of B lymphocytes with a production of anti-SARS-CoV-2 antibodies. (i) If the host is exposed to the SARS-CoV-2 virus, this vaccine-acquired immune response will control the infection, reducing the risk of developing coronavirus disease 2019 (COVID-19) and its most severe manifestations, including hospitalizations, intensive care unit admissions, and death related to COVID-19. The mechanisms involved in the development of myocarditis associated with anti-SARS-CoV-2 vaccination include the activation of innate immunity induced by exposure to viral nucleic acid (j) or exposure to the spike protein secreted by the host cell (k). In addition, this activation induces an exacerbated systemic immune response, including activation of natural killer (NK) lymphocytes, macrophages, and a massive release of cytokines that affect the cardiac muscle cells (l). Another mechanism is the generation of a T-helper-mediated immune response against autoantigens in cardiomyocytes (m). Finally, the presence of mimicry between the spike protein and cardiac autoantigens (e.g., myosin) generates anti-SARS-CoV-2 antibodies with affinity to cardiac proteins, inducing an autoimmune humoral response (n).

**Figure 4 viruses-13-02493-f004:**
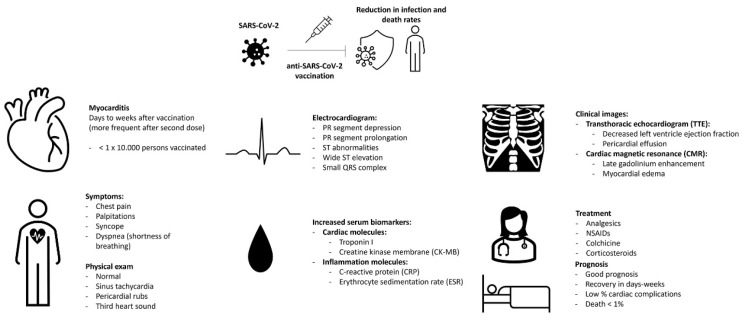
A visual summary of the epidemiology, clinical manifestations, laboratory, and treatment of myocarditis associated with anti-SARS-CoV-2 vaccination.

## Data Availability

Not applicable.

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
