# Peer review of "Cardiomyopathy Associated with Anti-SARS-CoV-2 Vaccination: What Do We Know?"

_viruses, 2021, doi:10.3390/v13122493_

Round 1

Reviewer 1 Report

  1. As the case report, the patients presented with myocarditis after vaccination and the authors described as the SARS-Cov-2 associated myocarditis. The authors described that it was vaccine induced myocarditis. Did the authors check other causes of myocarditis in this case?
  2. Some virus can also cause myocarditis and how does the author exclude the other causes in this case?
  3. The case was caused 10 days after second dose of administration. What is the possible mechanisms to describe as the vaccine induced myocarditis.
  4. Please add the possible mechanism at the revised manuscript.
  5. Regarding ethical issues, did the authors take consent for publication. Please describe briefly about consent at revised manuscript.

Reviewer 2 Report

This is a well written, very clear and interesting work reporting the known literature data on cardiomyopathies associated to anti-SARS-CoV-2 vaccination.

It could be published after the following revisions:

1-Title should be in lowercase letters not capital

2-It seems strange to me that a review starts with a Case Report and not with Introduction. Authors should consider to include the case report in a section following the Introduction. 

3-examples of disease reactivation after anti-COVID-19 (especially mRNA) vaccination could be briefly mentioned in the revised introduction citing works like: 10.1016/j.jdcr.2021.07.011 ; 10.3390/vaccines9091013 ; 10.3390/vaccines9060572 ; 10.1002/jmv.27036

4-more examples of literature works on available anti-COVID-19 vaccines should be added, for example authors could wish to cite after ref. 2-4 also the work with DOI: 10.2174/0929867328666210521164809

5-line 276 '...in soldiers, for example),..' add a reference 

6-I am more inclined to believe that the cause of the more frequent occurrence of cardiomyopathies associated to anti-SARS-CoV-2 vaccination with mRNA vaccines is due more likely to an immune response to the injected RNAs (even if made with modified nucleosides) than to the generation of autoantibodies (due to similarity between the encoded spike protein sequence and auto-antigens). In fact, the same spike is also produced after vaccination with adenovirus/DNA-carrying vaccines and the cardiomyopathic events should occur with the same frequence if the cause was the second.

7-are there any data published on cardiomyopathies (or other side effects) associated with inactivated anti-COVID-19 vaccines (such as the Sinopharm vaccine)? If yes, mention it briefly in the revised manuscript.

8-If figure 4 was taken from another published work, this should be cited and the permission for using the picture should be required if necessary. Otherwise, the figure should be remade by authors.

9-All acronyms should be explained at their first usage. See for example the legends of figures 2a and 2b

Reviewer 3 Report

The manuscript titled "Cardiomyopathy Associated with Anti-SARS-CoV-2 Vaccination: What do we know?" reports on cardiac involvement after vaccination with COVID vaccines. This topic continues to be of great importance as vaccination rates increase globally. This report is well written, thorough, and accessible to the greater audience. The figures are clear and easy to read. I offer some minor edits.

Line 82: Consider removing the comma after " good condition, with ..."

Line 100: Not sure by the meaning of "A little-described manifestation" . Consider revising to clarify. 
